# Performance Evaluation of Multi-Typed Precipitation Products for Agricultural Research in the Amur River Basin over the Sino–Russian Border Region

Yezhi Zhou [1,2], Juanle Wang [2,3,*], Elena Grigorieva [4], Kai Li [1,2] and Huanyu Xu [1]

1 College of Geoscience and Surveying Engineering, China University of Mining & Technology (Beijing), Beijing 100083, China
2 State Key Laboratory of Resources and Environmental Information System, Institute of Geographic Sciences and Natural Resources Research, Chinese Academy of Sciences, Beijing 100101, China
3 Jiangsu Center for Collaborative Innovation in Geographical Information Resource Development and Application, Nanjing 210023, China
4 Institute for Complex Analysis of Regional Problems, Far-Eastern Branch, Russian Academy of Sciences, 679016 Birobidzhan, Russia
* Correspondence: wangjl@igsnrr.ac.cn; Tel.: +86-64888016

**Abstract:** Precipitation data are crucial for research on agricultural production, vegetation growth, and other topics related to environmental resources and ecology. With an increasing number of multi-typed gridded precipitation products (PPs), it is important to validate the applicability of PPs and improve their subsequent monitoring capabilities to ensure accurate precipitation-based research. This study evaluates the performance of four mainstream PPs—European Centre for Medium-Range Weather Forecasts Reanalysis V5 (ERA5), ERA5-Land, Multi-Source Weighted-Ensemble Precipitation (MSWEP), and integrated multi-satellite retrievals for the Global Precipitation Mission (GPM)—in capturing the characteristics of precipitation intensity and derived agricultural drought in the crop-enrichment area over the Sino–Russian border region. The results show that, overall, GPM has the most balanced capability among the different experimental scenarios, with well-identified seasonal precipitation intensities. ERA5-Land had strong abilities in depicting annual distribution from spatial/stationary outcomes and obtained advantages in daily multi-parameter consistency verification. When evaluating monthly data in different agroclimatic areas, MSWEP and GPM had outstanding performances in the regions of Russia and China, respectively. For evaluating precipitation intensities and agricultural drought based on daily and monthly precipitation, MSWEP and GPM demonstrated finer performances based on combined agricultural thematic areas (ATAs). However, seasonal effects and affiliated material features were found to be the main factors in exhibiting identification capabilities under different scenarios. Despite good handling of intensity recognition in the eastern Chinese area, ERA5's capabilities need to be improved by extending sources for calibrating gauged data and information on dry–wet conditions. Overall, this study provides insight into the characterization of PP performances and supports optimal product selection for different applications.

**Keywords:** precipitation estimation; scenario analysis; intensity recognition; agricultural drought; Amur River Basin

## 1. Introduction

Characterizing the ability of precipitation products (PPs) to capture precipitation intensities has been a frequently studied topic in previous research [1–4]. Researchers and institutions have increasingly turned to estimating PPs using remote sensing and numerical simulation techniques, such as satellite inversion, model fusion, and multi-source assembly [5,6]. Various environmental and geographical factors directly impact the

suitability of the information utilization; hence, it is crucial for researchers to identify the applicability of the different products to obtain high-accuracy values. This uncertainty has a direct impact on agricultural regions, because less precipitation may lead to agricultural drought, and more precipitation may increase the flooding risk for crops. With increasingly extreme meteorological disasters and food-security challenges in the world, there is an urgent requirement for PP evaluation in cropland-rich areas.

Northeastern Asia (NA) is a crucial region for ensuring global food and health security, providing 2400–3000 Kcal of energy to the population in recent times. The region has also intensified farming activities to compensate for lost fields in other areas affected by natural disasters and political instability [7–9]. As a vital river system, the Argun, Amur-Heilong, and Ussuri flow along the Sino–Russian border and nourish the growth of abundant resources across the two countries. Furthermore, about 60% of the crops produced in this area are transferred to support global food security and Sustainable Development Goal 2 (SDG2 Zero Hunger) [10,11]. Given climate change effects in previous years, the spatio-temporal distribution of precipitation has led to significant changes in coping with the extreme development [12–14]; the aggravating floods and droughts caused by the condition have resulted in huge resource management and allocation losses in this area, especially crop production losses in the agricultural regions [15,16]. Due to the devastating hazards faced by this area, the capabilities to represent different classes of precipitation intensities and agricultural drought should be considered for the evaluation and selection of applicable PPs.

As sources of various satellite-based PPs, the Tropical Rainfall Measuring Mission (TRMM) and GPM were launched in 1997 and 2014, respectively; the estimations of their predecessors were reported to be enhanced in terms of light rain and spatial resolution [17] but to require further improvements in estimating the total amount of extreme precipitation [17,18]. Despite the satellite-based products providing relatively robust estimation, uncertainties arise when they are applied to high-altitude areas with perennial icy and snowy environments due to distorted radiation [19]. To address this issue, researchers have turned to numerical modeling for more accurate estimations. Detailed evaluation of the intensity patterns was enabled by the ERA-Interim, ERA5, ERA5-Land, and refined HAR datasets [19,20]. It was found that high-resolution datasets (ERA5-Land and HAR) have higher capability in dissecting the relationship of precipitation intensities to elevations in undulating regions. With the technologies' development, some inherent cognition of data applicability and maturity from the different typed PPs was gradually changed, and research on cross-type evaluation arose necessarily. For example, GPM was found to have generally higher performance in capturing extreme precipitation intensities, durations, and rates compared to ERA5, ERA5-Land, and GSMaP-Gauge. However, larger gaps in the three event indices were observed primarily in complex terrains for all the products.

Drought monitoring has become a critical concern in hydrological and meteorological studies. Compounded statistical models are increasingly utilized to compare drought monitoring capabilities to address the need for flexible time-scale datasets. With the comparison of atmospheric water deficits (AWDs), which were calculated using various elements collected by the stations, Zhu, et al. [21] evaluated the effectiveness of satellite-based precipitation products (SPPs) in monitoring drought patterns using the Standardized Precipitation Index (SPI) and found that GPM had the highest accuracy in the Xiang River Basin. To reduce parameter mismatches introduced by comparing multiple models, Rakhmatova, et al. [22] comprehensively tested the monitoring effectiveness of ERA Interim and ERA5 via the Standardized Precipitation Evapotranspiration Index (SPEI) in the arid region of Central Asia. Tigkas, et al. [23] proposed a modified version based on the original SPI, namely, the Agricultural Standardized Precipitation Index (aSPI), to obtain a finer performance in depicting the water amounts which can be used by crops for their productivities. The performance evaluation of drought magnitude was conducted using the comparison of SPI and aSPI, and the results showed that the latter is more robust than the original index. Keikhosravi-Kiany, et al. [24] employed a "grid to grid" method with upscaling

of in situ observations to satellite-derived grid scales and comprehensively examined the performance of TRMM3B43, GPM, CHIRPS, and ERA5 for drought assessment in Iran. The evaluation results showed that TRMM3B43 and GPM were preferentially recommended at both short and long time scales. Given the cross-generational and urgent transfer phenomena between floods and droughts, Bai, et al. [25] studied the performance of extreme precipitation events (EPEs) captured by satellite-retrieved reanalysis and blended multi-source datasets at multi-temporal scales. Indices with days on which daily precipitation was <1 mm (dry days) were underestimated by these datasets, while the corresponding results were mostly overestimated for the indices characterizing the situation of recording the heavy precipitation. According to such research and other homogeneous studies acting in various regions [1,26–28], the evaluation results of precipitation intensities and drought events are independently provided, though the internal links between the two still need to be evaluated and analyzed comprehensively. Regarding the other aspects of agricultural regionalization, the previous research evaluating PP estimations for agricultural issues has classified the districts based on resources, such as temperature, precipitation, and soil characteristics [9]. However, further investigation is needed to achieve a finer distribution of specific subregions.

The objectives of this study are: (1) to comprehensively understand the error characteristics of precipitation estimates through an intercomparison of different datasets against gauge data collected over a recent 5-year period (2015–2019); (2) to define combined Sino–Russian agricultural thematic areas (ATAs) and analyze the distribution and frequency of precipitation intensities and agricultural drought characterization within these areas; (3) to provide scientific references for combined hydrological management and meteorological disaster prevention in this high-latitude transboundary region. The remaining sections of this paper are organized as follows: the selected precipitation products and evaluation metrics will be described in Section 2, followed by an analysis of the assessment results in Section 3. Finally, the discussion and conclusion will be presented in Sections 4 and 5, respectively, with the aim of providing scientific references for hydrological management and meteorological disaster prevention in this high-latitude transboundary region.

## 2. Materials and Methods

### 2.1. Study Area

The study area is in the cross-border region between Northeastern China and the Russian Far East, located in the Amur River Basin, which comprises seven administrative units, including Heilongjiang Province, Jilin Province, Liaoning Province, Inner Mongolia, Amur Oblast, Jewish Autonomous Oblast, and Primorsky Krai. This region is considered the main agricultural production area in Northeastern Asia, is rich in natural resources, and is characterized by favorable geographical and climatic conditions and potential for the development of agricultural resources [7]. The climate in the eastern part of Northeastern China is characterized by a temperate continental monsoon, while the western region is dominated by a continental climate [29,30].

Beginning in the 21st century, the average temperatures of the Chinese section remained at approximately 4 °C, with an average precipitation of 250–800 mm. The Xiliaohe, Songnen, and Sanjiang plains are located in the western, middle, and eastern regions of the area. These regions are fertile, supporting the cultivation of crops, such as soybeans, rice, corn, and potatoes. Planting crops in the Russian Far East is affected by environmental factors, such as cold temperatures and permafrost. Hence, planting has mainly been concentrated in the southern and eastern administrative regions with better natural conditions. The Zeya–Bureya and Khanka Lake plains, located in southern Amur Oblast and western Primorsky Krai, are the key agricultural areas. The rain and heat conditions of these fields are suitable for cultivating soybeans, rice, and cereals (such as barley, oats, and buckwheat), while fruit planting is also being vigorously developed in the Jewish Autonomous Oblast [31].

To test the effectiveness of different PPs on agriculture, the study area was divided into ten agroclimatic regions (Figure 1) by the given patterns of "the regionalization of agro-climate in China" and "Agroclimatic Zoning of Russia", as proposed in the Chinese and Russian literatures on the consideration of the Chinese geographical situation and economic development (C-1: the North Greater Khingan farming region; C-2: the East Inner Mongolia farming region; C-3: the Songliao farming region; C-4: the Lesser Khingan/Mount Changbai farming region; C-5: the Sanjiang farming region) and Russian annual effective accumulated temperature ($\sum ET_{10}$) (R-1: $\sum ET_{10} < 1000$ °C; R-2: 1000 °C < $\sum ET_{10} < 1100$ °C; R-3: 1100 °C < $\sum ET_{10} < 1200$ °C; R-4: 1200 °C < $\sum ET_{10} < 1300$ °C; R-5: $\sum ET_{10} > 1300$ °C) under the current anthropogenic mobility and climate change [32–34].

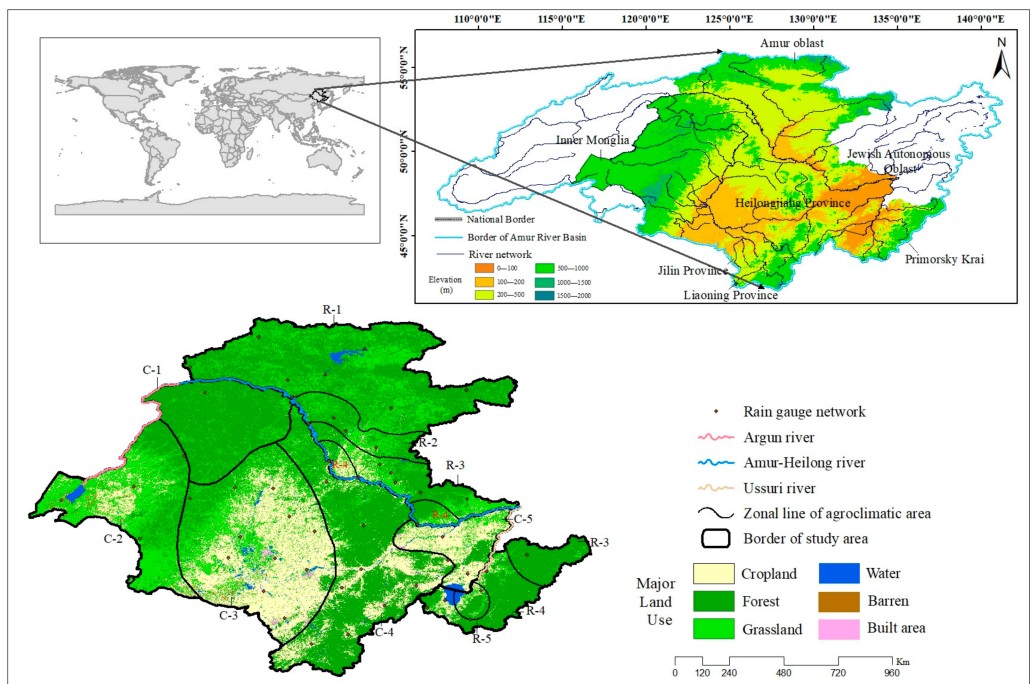

**Figure 1.** Study area.

*2.2. Data*

2.2.1. Data Sources

Four globally scaled PPs (ERA5, ERA5-Land, GPM, and MSWEP), which are derived from different types and have been shown to perform well in previous research, were selected to test their applicability. ERA5 and ERA5-Land afforded detailed surface variables and provided a relatively higher spatial resolution compared to other reanalysis datasets [35]. Both products are available online through the Copernicus Climate Change Service (C3S) Climate Data Store (https://doi.org/10.24381/cds.adbb2d47 and https://doi.org/10.24381/cds.e2161bac, accessed on 11 October 2022). GPM is a global multi-satellite precipitation product. The sixth and final edition of the product was used in this study and can be obtained online at https://disc.gsfc.nasa.gov/datasets/GPM_3IMERGDF_06/summary, accessed on 12 October 2022. MSWEP is a long-term, global-scale precipitation product that can be obtained online at http://www.gloh2o.org/mswep/, accessed on 13 October 2022. This product takes advantage of the strengths of gauge, satellite, and reanalysis data. MSWEP v2.8 was used in this study. Detailed information about the selected PPs listed above has been provided in Table 1.

**Table 1.** List of global-scale PPs used in this study [35–38].

| Product Name (Abbr.) | Full Name | Product Principle | Spatial Resolution | Temporal Resolution | Cover Time |
|---|---|---|---|---|---|
| ERA5 | The fifth generation European Centre for Medium-Range Weather Forecasts Reanalysis | Reanalysis | $0.25° \times 0.25°$ | Hourly | 1950 to present |
| ERA5-Land | The fifth generation European Centre for Medium-Range Weather Forecasts Reanalysis on global land surface | Reanalysis | $0.1° \times 0.1°$ | Hourly | 1950 to present |
| MSWEPv2.8 | Multi-source weighted-ensemble precipitation | Multisource- based | $0.1° \times 0.1°$ | 3-hourly | 1979 to present |
| GPM_3IMERGDF | Integrated multi-satellite retrievals for global precipitation measurement Level 3 (1-day V06) | Satellite-based | $0.1° \times 0.1°$ | Daily | 2000 to present |

The daily rain-gauge observations from 2015 to 2019 were collected from the dataset "Global Summary of the Day" (GSOD) (https://www.ncei.noaa.gov/data/global-summary-of-the-day/archive/, accessed on 14 October 2022) provided by the National Oceanic and Atmospheric Administration. Then, the data were rechecked and replenished by the correlated datasets provided by the national meteorological data service centers of China and Russia (http://data.cma.cn and http://aisori-m.meteo.ru, accessed on 14 October 2022). The spatial distribution of the rain-gauge network over the study area is shown in Figure S1. Before using these raw data for evaluation, the datasets were tested with three progressive levels of quality controls, including deleting the stations with unavailable records (missing or without quality control) >100 days, throwing off anomalous observation values, and checking the spatial/internal consistency of the data. Finally, the data from 50 stations (28 in Chinese and 22 in Russian regions) that were covered by all the PPs were screened and used in the evaluation process.

### 2.2.2. Data Preprocessing

For the precipitation products using the estimations by ERA5, MSWEP, and GPM, the first two-hourly series were accumulated for the daily scale and all the products were extracted and converted into original netCDF files in GeoTIFF format using the corresponding Python packages, such as numpy, datetime, and gdal. Considering the fact that the in situ station data were sparse and unevenly distributed, upscaling the observations to grid scales would have produced unreliable results in the ungauged region. Therefore, we performed multi-value extraction from the grids to the locations of the selected stations using the bilinear interpolation method and evaluated the applicability of the data by comparing them with the observed data (grid to point). Since the datasets had different spatiotemporal resolutions, the large differences in resolution would have caused biases in the data values obtained through the interpolation. Furthermore, the precipitation evaluation on complex terrains, such as that of Greater/Lesser Khingan in the study region, would introduce uncertainty; thus, we used the cdo remapbil package to regrid ERA5 into 0.1°. The GPM dataset supplied two types of daily-scale results that were estimated by multi-satellites (combined with microwave-infrared (IR)) and all available microwave (MW) sources. The extracted data were primarily evaluated by the rain-gauge observations, and the values with the lowest differences from the observations were chosen to represent the estimations as the final daily data. For the ERA5-Land product, as the data recorded at the hourly scale were accumulated rather than instantaneous values, the first data for a certain day were extracted by the Climate Data Store (CDS) toolbox (https://cds.climate.copernicus.eu/toolbox-editor/, accessed on 14 October 2022) and recorded as the daily values of the previous day. Finally, the data of four PPs were employed to test their consistency at multi-temporal scales through the resample method.

*2.3. Methodologies*

2.3.1. Consistency Verification of the Selected Products against the Gauge Data

In order to comprehensively evaluate the products' quality of recorded precipitation, we used different statistical indices to depict the consistency of the results between the estimations and gauge values. Among the indices, the correlation coefficient (CC) is the measure of the strength of the relationship between the relative movements of two variables. CCs generally reflect the strength of the relationships between estimated and observed datasets. RMSE and BIAS are common scores used to describe the degrees of deviation between estimated and observed datasets.

2.3.2. Evaluation of the Characterization of Precipitation Intensity

The classes of precipitation intensity according to the national standard document (GB/T 28592-2012) are listed below (Table 2), which were formulated by the Chinese meteorological administration (CMA).

**Table 2.** Classification of precipitation intensities [39].

| Precipitation Amount in 24 h (mm) | Precipitation Intensity Class |
|---|---|
| <10 | Light precipitation |
| 10–25 | Moderate precipitation |
| 25–50 | Heavy precipitation |
| 50–100 | Downpour |
| 100–250 | Torrential precipitation |

To assess the performance of the intensity identifications obtained by the PPs, three evaluation indices (Table 3) were derived from the elements of the fusion matrix.

**Table 3.** Indices for evaluating the accuracy of characterizing precipitation intensity.

| Name | Equation | Best Value |
|---|---|---|
| Probability of Detection (POD) | $\frac{H}{H+M}$ | 1 |
| False-Alarm Ratio (FAR) | $\frac{F}{H+F}$ | 0 |
| Equitable Threat Score (ETS) | $\frac{H-H_e}{H+M+F-H_e}$ <br> $H_e = \frac{(H+F)(H+M)}{H+F+M+C}$ | 1 |

Notes: *H*, hit case, refers to cases where both observations and estimates are greater than or equal to the designed threshold; *M*, missed event, refers to cases where observations are greater than or equal to the designed threshold but where estimates are less than the threshold; *F*, false alarm, refers to cases where estimates are greater than or equal to the designed threshold but observations are less than the threshold; *C*, correct negative, refers to cases where both observations and estimates are less than the designed threshold.

2.3.3. Assessing Ability to Identify Agricultural Drought Events

Agricultural droughts directly act on the continuous shortage of soil water, affecting the normal growth and development of crops. To add parameters expressing the soil water balance and the precipitation amount which can be stored in the vegetation, Tigkas, Vangelis and Tsakiris [23] designed the agriculture-oriented Standardized Precipitation Index (aSPI) by introducing the concept of effective precipitation ($P_e$) (Equation (1)) into the Standardized Precipitation Index (SPI) which was proposed by Mckee, et al. [40].

$$P_e = 25.4 * SF * \left( 0.04931 * P^{0.82416} - 0.11565 \right) * 10^{0.000955 * ET_c}, \tag{1}$$

$$aSPI = -\left( t - \frac{C_0 + C_1 t + C_2 t^2}{1 + d_1 t + d_2 t^2 + d_3 t^3} \right), 0 < H(x) \leq 0.5 \tag{2}$$

$$aSPI = t - \frac{C_0 + C_1t + C_2t^2}{1 + d_1t + d_2t^2 + d_3t^3}, 0.5 < H(x) \leq 1, \tag{3}$$

where $t = P_e/\beta$, $P$ is the precipitation value, and the meanings of the other elements can be found in the literature [23].

Run theory is a proven method for analyzing time-series data to identify droughts [41]. It can indicate the processes of drought events by determining values of the selected drought evaluation index.

## 3. Results

### 3.1. Accuracy Evaluation of Records from Selected Precipitation Products

#### 3.1.1. General Performance

Comprehending the overall precipitation patterns in the cross-border region is crucial for evaluating the performance of precipitation products and understanding their spatial distribution. In this section, the mean annual precipitation levels obtained by PPs and gauge data are shown in Figure 2.

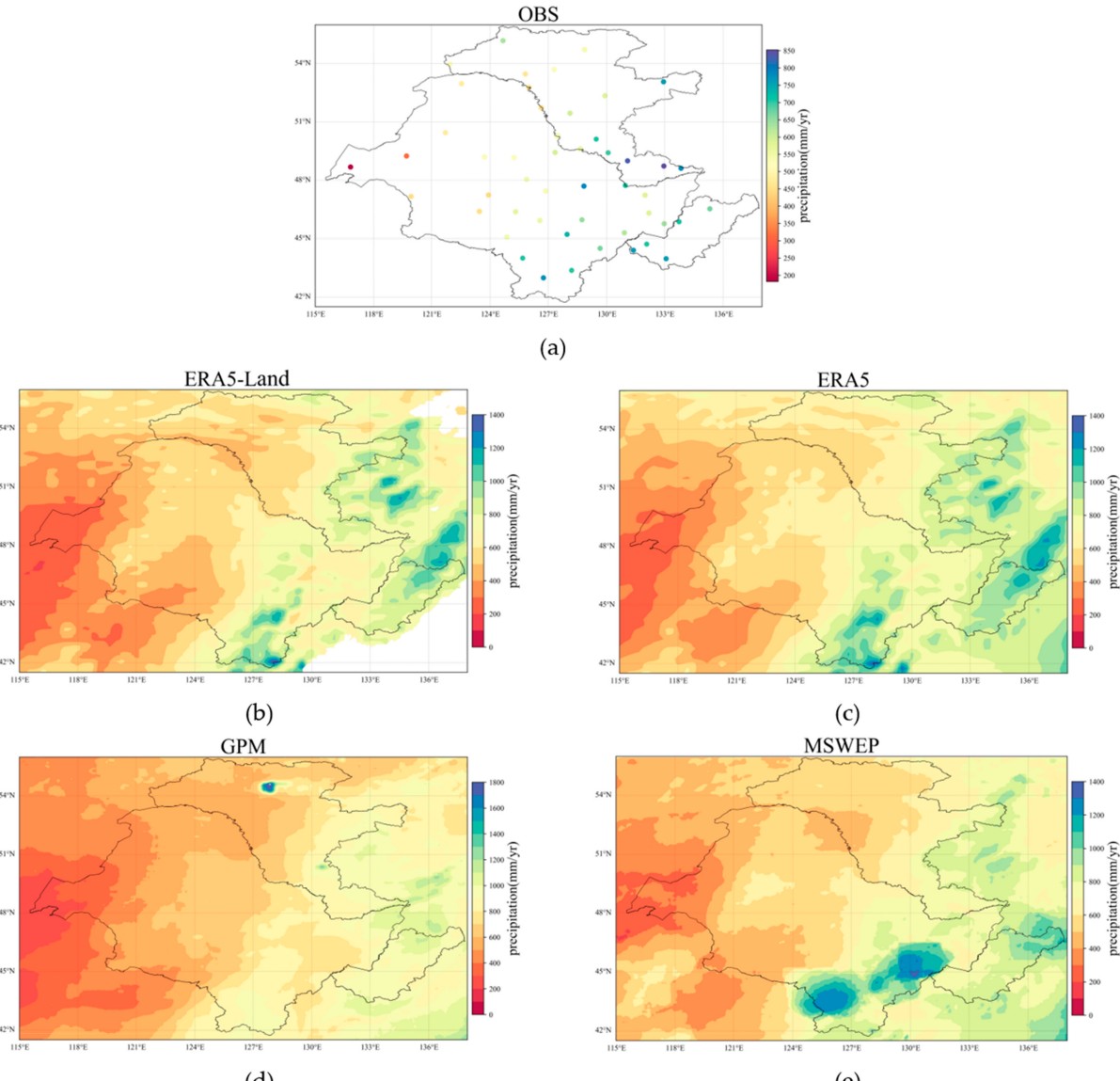

**Figure 2.** Spatial distribution of the mean annual precipitation over the study area: (**a**) rain-gauge observations (OBS); (**b**) ERA5-Land; (**c**) ERA5; (**d**) GPM; (**e**) MSWEP.

The mean annual precipitation values from all datasets shown in the figure reveal a spatial pattern of "high in the southeast and low in the west". The lowest precipitation levels (<200 mm) occur at the edge of the western region, while the highest levels (1200–1400 mm) are generally observed in the southern Chinese and eastern Russian regions. Among the four selected PPs, ERA5 and ERA5-Land exhibit relatively higher similarity levels compared to the other products. However, compared to the gauge data, the underestimation of precipitation from these two products is mainly distributed in the eastern Chinese region. Conversely, they jointly overestimate precipitation in the southern and eastern regions of the study area. In addition, ERA5 exhibits distinctive overestimation in the central region of the Chinese section. In terms of the estimation using MSWEP, precipitation in the southern border area is highly overestimated, and the estimation from GPM along the Zeya Reservoir is also overestimated. Despite these abnormal regions, both MSWEP and GPM produce relatively reasonable estimations. Overall, ERA5-Land provides the best performance compared to the other products. However, the corresponding results from MSWEP and GPM are better in terms of depicting precipitation in the northern and southern regions, respectively. Figure 4 represents the annual precipitation levels for each station from 2015 to 2019 to show the consistency of different PPs with the observations:

The annual precipitation distributions of four PPs (Figure 3) exhibited high consistency with the gauge data (r (CC) > 0.75). The general trends revealed varying degrees of overestimation with the different PPs, and these phenomena gradually weakened with the increase in precipitation amounts (there was even a slight underestimation in the higher range of precipitation from GPM). The high overestimation (biases over 300 mm/yr) status of the four PPs was gathered in the gauge data range between 600 and 900 mm/yr, while the underestimation distribution was relatively flat, especially in the ERA series. The annual results from ERA5-Land had the highest consistency with the gauge data and performed especially well in estimating the areas with low precipitation.

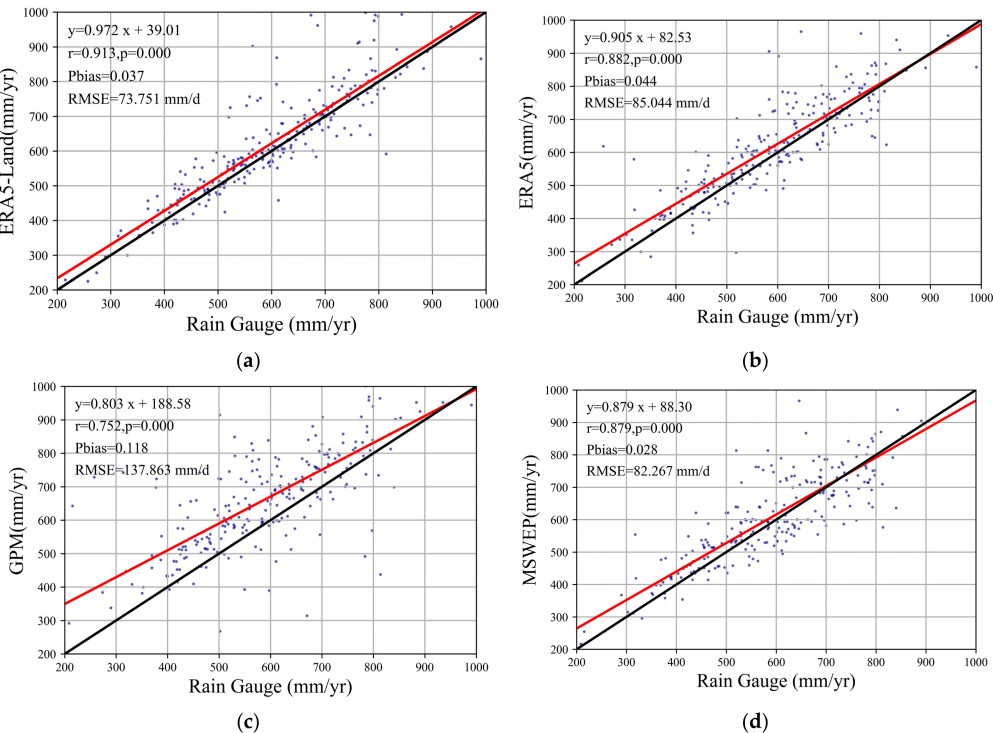

**Figure 3.** Scatter diagram of annual precipitation distribution observations plotted against results from PPs: (**a**) ERA5-Land; (**b**) ERA5; (**c**) GPM; (**d**) MSWEP. The black lines are the 1:1 lines, and the red lines indicate the regression lines between the rain gauges and PPs.

### 3.1.2. Parameter Evaluation

Figure 4 shows the spatial patterns of performances for consistency between the daily PP and rain-gauge data. In the Chinese section, the four PPs generally exhibited higher CCs in the southern region, while they had low performances (CC < 0.4) in the west. The ERA5-Land product data had greater consistency with the gauge data; the consistencies of MSWEP and GPM were basically the same, while the weaknesses were in the northern and southeastern regions, respectively. ERA5 had the poorest representation in the cross-border region compared with others. For the Russian side, MSWEP had the best performance for all stations, especially in the eastern region. For the corresponding conditions of ERA5-Land and GPM, all the results fell in the range of 0.5–0.7, while the consistency conditions of ERA5-Land were generally better than those with GPM. The consistencies of most results obtained by ERA5 were unsatisfactory among the Russian stations.

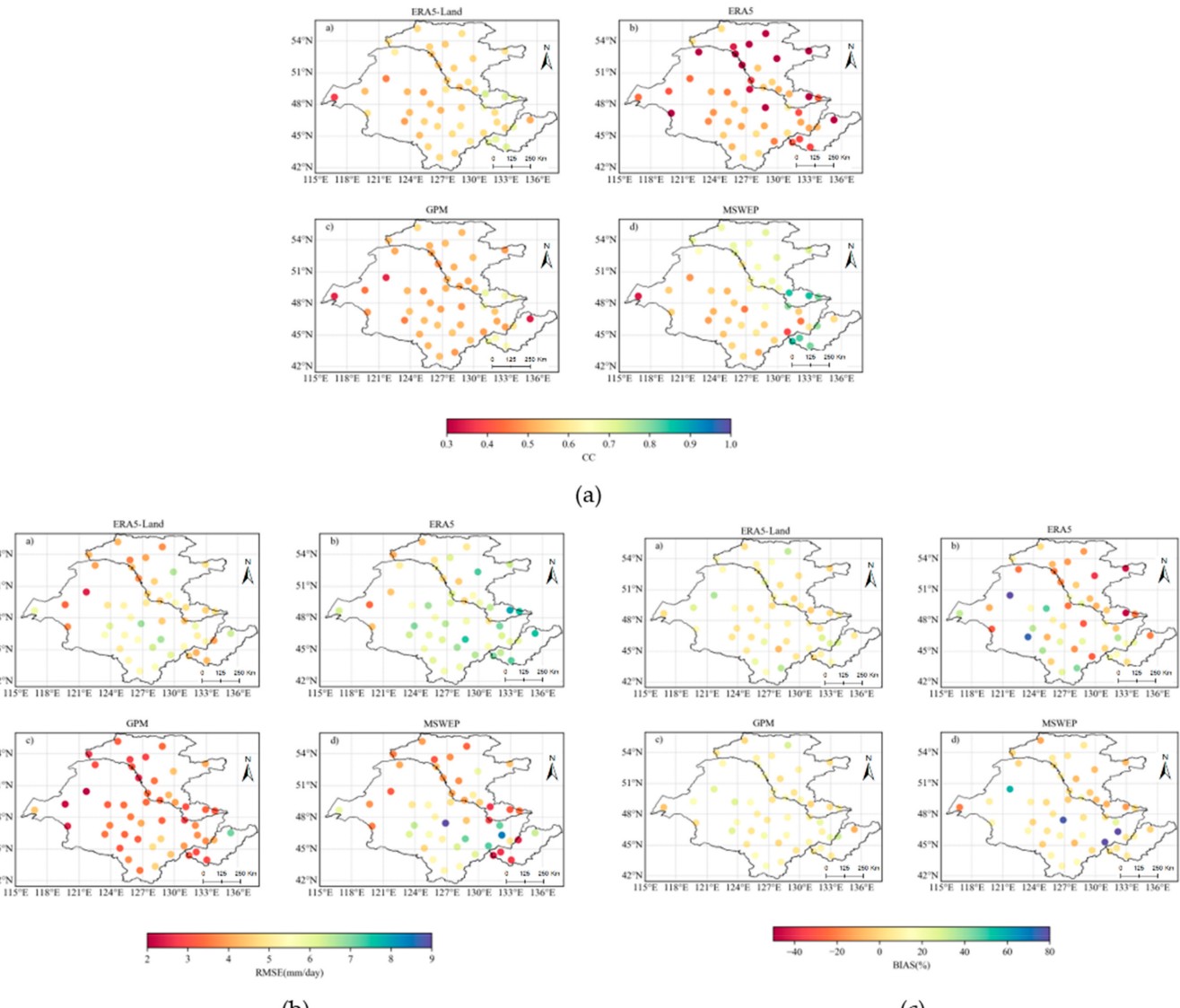

**Figure 4.** Spatial patterns of performances in terms of the statistical indices for consistency from PP estimations and observations at the daily scale from 2015 to 2019: (**a**) CC; (**b**) RMSE; (**c**) BIAS.

Based on the range between 2 and 9 mm/day, of all the PPs, GPM had the best performance for RMSE, with the general distribution pattern of deviation increasing from west to east. The estimation accuracy from MSWEP was closest to that of GPM on the Russian side, while in the Chinese section, the secondary acceptable outcome was generated by ERA5-

Land. For the four products, certain regions of high deviation existed in the central region of the study area, especially on the border between Heilongjiang Province and Primorsky Krai. In quantitively examining the evaluation indices of estimation conditions, the biases between the four PPs and the gauge data fell in the range of [−50% (underestimation), 80% (overestimation)], the relatively slight deviation (the absolute value of BIAS < 20%) occurred in the central region as per the results from ERA5-Land and GPM. The results of BIAS were alleviated in the MSWEP estimations, except for the Chinese stations located beside the eastern regions of Russia.

Monthly precipitation (MP) was adapted as a suitable temporal scale and widely used to evaluate the climatic situation for agricultural production in certain regions. To test the oriented performances in estimating the MPs of different agroclimatic regions, the data were collected and sorted by stations in the same district. The evaluation results are shown in Figure 5.

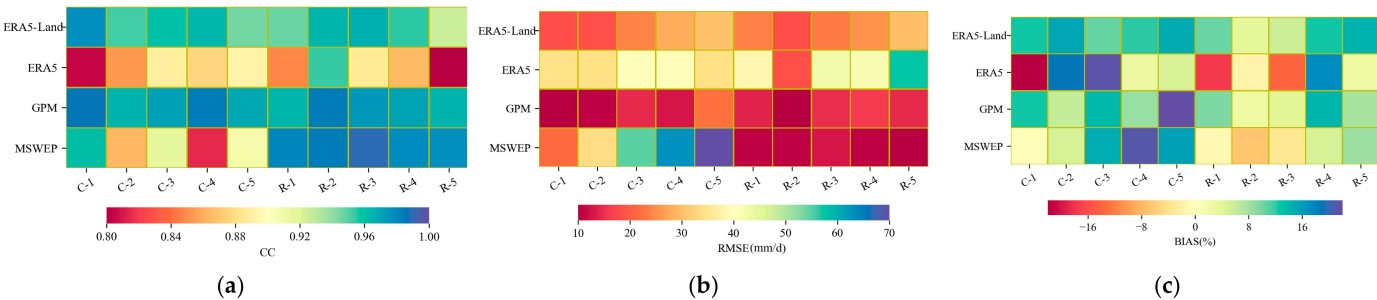

**Figure 5.** Performances in terms of statistical indices of consistency estimated by PPs among the different agroclimatic areas from 2015 to 2019: (**a**) CC; (**b**) RMSE; (**c**) BIAS.

In Figure 5, the consistencies for each agroclimatic area with the PPs can be seen. On the Chinese side, the estimation from GPM was recommended, while the more convincing results turned out to be those provided by MSWEP on the Russian side when considering CCs. The recommendations remained the same for RMSEs. The BIAS calculated by the MSWEP-gauge combination obtained a better response in the northwest region (C-1/C-2). ERA5-Land and ERA5 were the ideal selections for assessment of the middle (C-3) and eastern (C-4/C-5) zones, respectively. For Russian agroclimatic areas, all the products performed well (BIAS values < 10%) in the middle part of Amur Oblast (R-2) and Primorsky Krai (R-5). In the rest of the area, MSWEP can be seen as the superior product to be applied in research dissecting the association between precipitation and related agricultural transactions.

### 3.2. Performance Evaluation Regarding the Derived Precipitation Events in the Agricultural Thematic Areas (ATAs)

In practical applications, information from different PPs on the identification accuracy of precipitation intensity and derived drought events will be a more meaningful reference, especially in conducting correlation research when studying the impact of precipitation amount on agricultural production. In the following section, several agricultural thematic areas (Table 4) were set based on the previous agroclimatic division considering geographic location, farmland range, and rain-gauge network intensity. Then, the two aspects of evaluation were conducted among these areas.

**Table 4.** Components of Sino–Russian agricultural thematic areas (ATAs).

| Country | Name | Related Agroclimatic Region | Agricultural Characteristics | Total Number of Stations |
|---|---|---|---|---|
| China | C-I | C-1/C-2 | The region has the distinctive features of farming–forestry/pastoral interlacing and providing abundant resources, including forestry, fruit, cash crops, as well as the husbandry industry. | 5 |
|  | C-II | C-3 | The middle part of Songnen Plain is the core region of Chinese farming produce, mainly producing staple grains, such as maize, rice, and wheat. | 10 |
|  | C-III | C-4/C-5 | With the spatial pattern of "two mountains (Lesser Khingan and Mount Changbai) nip one plain (Sanjiang)", the region contains different cultivated types of grain, fruit, and aquatic products. | 13 |
| Russia | R-I | R-3/R-4 (in Amur Oblast and Jewish Autonomous Region) | Having the longest history of crop production in the southern part of the Russian Far East, the soybean and grain produced on the vast Amur–Zeya Plain were two major planting types in this area. | 8 |
|  | R-II | R-3/R-4/R-5 (in Primorsky Krai) | With adequate water resources and a humid monsoon climate, the major planting industry is concentrated in the edge area surrounding Khanka Lake. In recent years, the increasing pattern of cropland in this area was observed with the expansion of the production of the agricultural sector [42]. | 5 |

3.2.1. Evaluation of the Distribution and Frequency of Precipitation Intensity

To investigate the evolutional laws of precipitation intensity, the daily precipitation levels of the stations in the ATAs were sorted by different seasons to acquire their distinctive features during the four periods. The seasons of spring, summer, autumn, and winter are defined as March–May (MAM), June–August (JJA), September–November (SON), and December–February (DJF), respectively.

The left subplots in Figure 6 indicate that the four precipitation products generally underestimated heavy rainfall intensities and overestimated relatively light rainfall intensities. The phenomenon of underestimation was least apparent in the case of GPM. During spring and autumn, the maximum range of daily precipitation estimated by these products was 40–60 mm. Light precipitation was more highly overestimated during autumn than during spring. In summer, the more intense levels extended further and the remaining individual records were over 100 mm/d and were underestimated. During winter, intensities decreased and fell in the range of 0–20 mm/d. ERA5-Land was the best-performing product in estimating precipitation during this season, while MESWP estimation had relatively large biases against the gauge.

According to the right plots in Figure 6, all four products first overestimated and then underestimated the corresponding frequencies of precipitation intensities during summer. The intersecting point of JJA_PE and JJA_OBS occurred in the range of 20–25 mm/d, indicating the separation of the differences between the two types of frequencies. The rank of biases ranged from large to small: ERA5-Land, ERA5, GPM, and MSWEP. The results for the spring and autumn seasons had similar patterns, with small biases when compared with observations. For the former bins of precipitation intensities, the frequencies were much higher in autumn than in spring. During winter, the corresponding precipitation intensity was normal, in the range of ~20 mm/d, and the deviation condition against the

observation was relatively controlled by the ERA series. To further visualize the general identification accuracies in the different ATAs, the multi-class intensities of the four PPs from 2015 to 2019 are depicted in Figure 7.

In a general view of all the regions (Figure 7a), characterized by "hook" shapes, the intensity classes below 10 mm/d decreased first and then increased. In the middle class of "moderate precipitation", the divisions were fundamentally fair and occupied about 10% of the total intensities. In the intensity classes of "heavy" or above, referring to the range of [50, 250], the precipitation bins were rare in these blocks, and the precipitation intensities were briefly underestimated among the four selections. All the PPs in the present study were competent for capturing the patterns of intensity for daily precipitation. Figure 7a shows that in the classes of" light rain" (0–10 mm/d), there was overestimation under the condition of intensity <2 mm, and this is more obvious in the former bin, while underestimation was recorded in the rest of the scopes and gradually increased in severity. To describe the differences, the figure and its subplot show that MSWEP was preferred for estimating most cases, while GPM and ERA5-Land highly underestimated the intensities in the range of light and heavy precipitation, respectively.

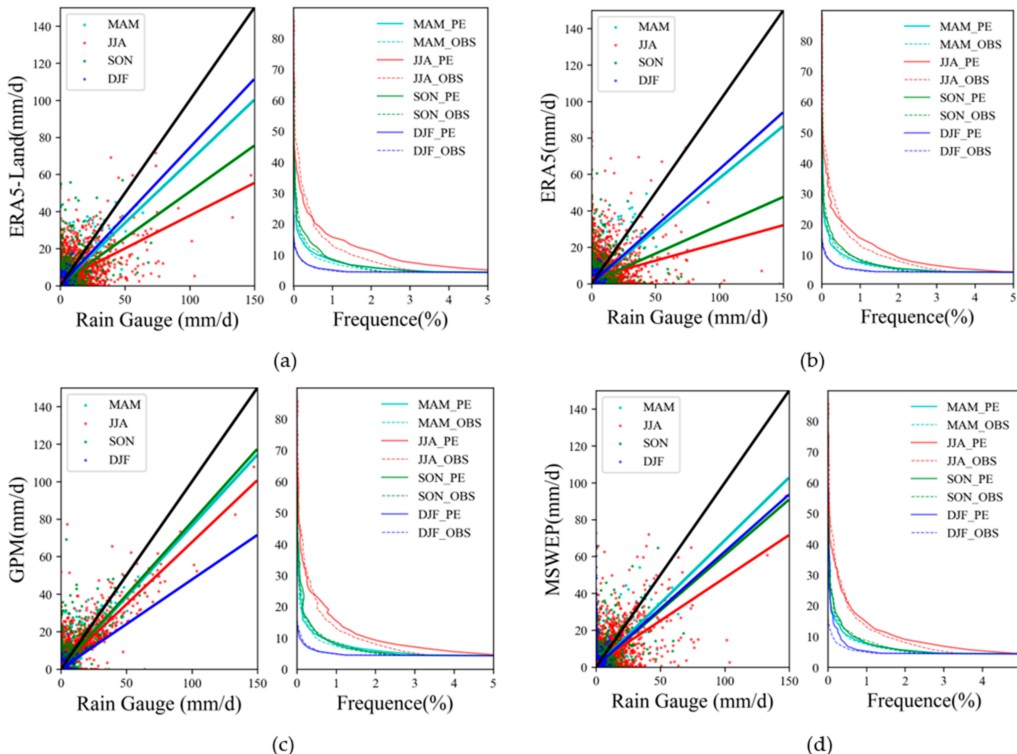

**Figure 6.** Statistical characteristics of different frequencies based on the intensities of four PPs and rain-gauge observations in the four seasons: (**a**) ERA5-Land; (**b**) ERA5; (**c**) GPM; (**d**) MSWEP. The left subplot indicates the seasonal scatter plots between rain-gauge observations and PPs. The right subplot indicates the seasonal frequency distribution of rain-gauge observations (OBS) and PPs (PE). The black lines in the left subplots are the 1:1 lines, and the other lines (the colors were set according to the same scheme as the seasonal plots) indicate the regression lines between the rain gauges and PPs.

For the probability distributions of intensities in each basin, the patterns were basically in accord with those of the assembled region. On the Chinese side, the C-I area had higher ratios of microscale precipitation (<1 mm), with a corresponding decrease in the range of "moderate precipitation" when compared with other parts. ERA5 products dominated, except for the initial class. For the regions of C-II and C-III, in the range of "slight rain", the condition of intensities in C-II were generally underestimated, while the overestimation occurred in most bins in the C-III region. For intensities in the 10–25 mm range, GPM and MSWEP had the best performances in estimating this type of precipitation. In terms of

"heavy precipitation", MSWEP took the lead in simulating the observed data. As for the condition of the Russian side (R-I and R-II), ERA5 products turned out to be highly underestimated in the setting of intervals in the R-I region. With the increase in strength, MSWEP products gradually emerged as advantageous in precipitation estimation, particularly in the range of 15–20 mm. For the corresponding condition of R-II, the differences in the deviations between the four products were lower than those in other scenarios, while GPM and MSWEP preceded obviously in the comparison when set in the "heavy" precipitation intensities.

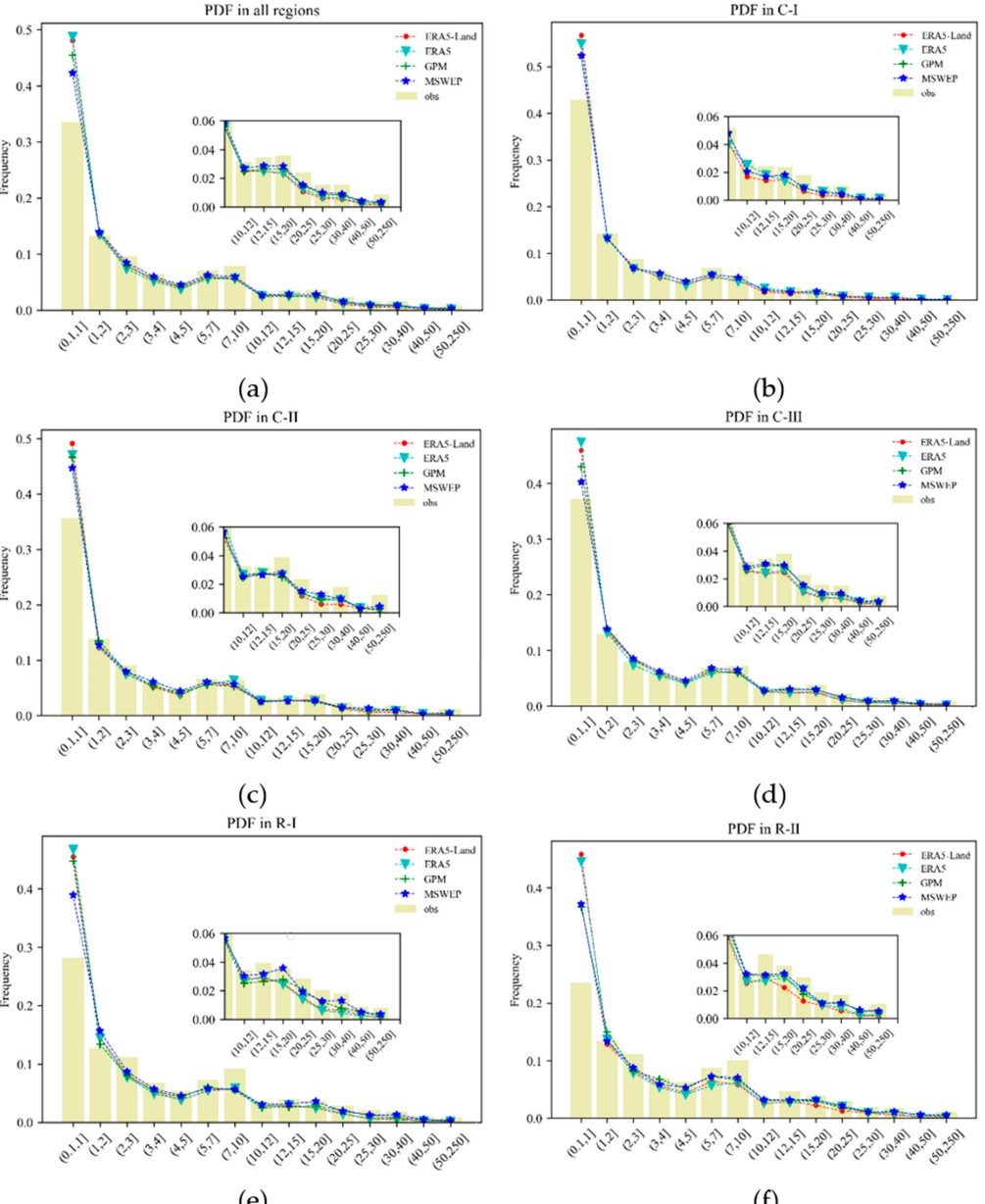

**Figure 7.** Probability density functions (PDFs) of precipitation intensities given by PPs and observations (obs) in each ATA from 2015 to 2019: (**a**) all regions; (**b**) C-I; (**c**) C-II; (**d**) C-III; (**e**) R-I; (**f**) R-II.

To quantitatively evaluate the accuracy of the data from ATAs which belonged to both the Chinese and the Russian parts, the "one-to-one" daily precipitation from the PPs and rain-gauge data was marked by precipitation intensities ranging from "slight" to "heavy" in steps of 0.5 mm/h. Then, the cases introduced in Table 3 were combined and calculated. The assessed results are shown in Figure S1.

The top subplots of the figure show a descending pattern of POD values in both China and Russia, with a gradually decreasing rate of variation. In the initial region, the highest score of "0.5 mm/d" was 0.8, but for most selections, the score sharply decreased by 0.5 as the precipitation intensities approached 5.5 mm/d. The trend then slowed down as the score of 0.3 was obtained around the intensity of 15.5 mm/d. The data for Russian ATAs exhibited some degree of fluctuation, and more PPs had identifications than those in the Chinese regions. In terms of product evaluation, MSWEP and GPM were found to be the best products for precipitation estimation, while ERA5 ranked last in both regions. To assess the models' abilities to predict the purity of positive samples correctly, the results evaluated by FAR indicated that the scores for ERA5, ERA5-Land, and MSWEP in the Chinese ATAs gradually increased from 0.5 to 0.85, while GPM showed a slight decrease in the range of [0.35, 0.45]. The biases of the four PPs were relatively alleviated in the Russian regions.

Earlier studies have proven that excessive or deficient samples of "hits" and "false alarm" events for specific thresholds may affect the accuracy of binary-constructed indicators [27,43]. Considering more elements, the skill scores of ETS were used to test the PPs' accuracy comprehensively. The corresponding results (Figure 8) distinguished the selections among the preliminary thresholds more clearly, while the evaluation changes were not evidently revealed in the last sections. According to the above three indices at the different thresholds, all PPs did not perform well in capturing different amounts of precipitation, and their performances degraded with the thresholds' expansion, implying that the PPs struggled to accurately capture heavy precipitation at the daily scale. In general, all the scoring indicators showed that GPM had superior performance relative to the other PPs for all the regions, while the ERA5 product lagged in these evaluation processes.

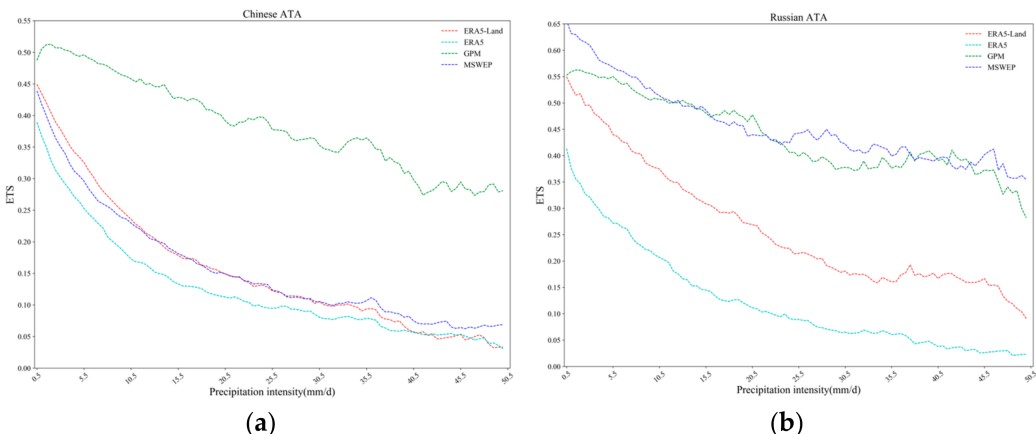

(a)　　　　　　　　　　　　　　　　(b)

**Figure 8.** Assessment of different precipitation intensities among the Chinese (**a**) and Russian (**b**) ATAs recorded by PPs based on the total factors of the fusion matrix.

3.2.2. Evaluation of Agricultural Drought Monitoring Effectiveness

According to the methods introduced in Section 2.3.3, MPs from the selected stations were employed to calculate the aSPIs. Among the different timing scales of the index, aSPI-1 values (obtained per month) were selected for the results. In assessing the capturing ability of the different drought event classes, the threshold (aSPI = −0.5) which represented "slight" drought in Table 3 was set to count the characterizations and compare the identifying feature identifications from different PPs. The corresponding performances with respect to the drought index and events are shown in Figure S2 and Figure 9.

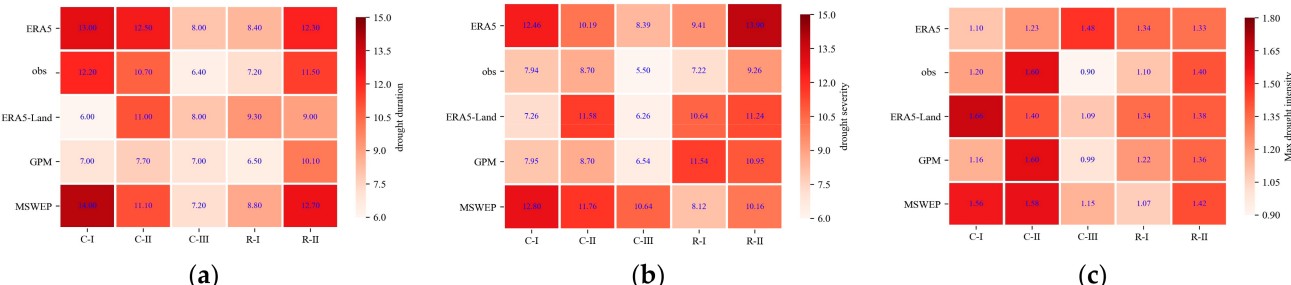

**Figure 9.** Drought event characterization given by the primary data collected by the meteorological stations (monthly precipitation value) and the PPs using the aSPI and run theory from 2015 to 2019 in the ATAs: (**a**) average duration; (**b**) average severity; (**c**) average max intensity.

Figure S2 shows that the general condition of most PPs showed good agreement with the observations and that dry–wet conditions showed periodicity variation with large amplitudes of oscillation. Evaluations of heavy drought (aSPI < −1.5) were likely to appear in the months during summer and autumn, becoming more frequent especially at the start and end of these seasons, respectively. The monthly indices generally showed a wetter trend in recent years. For the two Russian ATAs, drought and flood events tended to occur at the transition between spring and winter, and, compared with the rest of the products, MSWEP and GPM generally fitted with these observations, respectively. In contrast, ERA5 obviously overestimated wet conditions in the eastern Russian region (R-II) during 2017–2019. For the conditions among the corresponding regions on the Chinese side, the first two areas (C-I and C-II), with relatively similar variation in rises and falls, GPM data showed, overall, high consistency with the observations. For the region of C-III, the MSWEP estimation led to capture of the monthly evaluation of dry–wet identifications, while in the Chinese areas, the degree of deviation between the calculations of the ERA5 product and the measured data was the highest among the selections.

Given that aSPI-1 can better reflect the degree of dry–wet abruption, the study evaluated the identification abilities of agricultural drought (Figure 9). Using run theory, three average indicators were obtained for each thematic area over the past five years. The results showed that the agricultural drought durations monitored by the aSPI index based on GPM and ERA5-Land were seriously underestimated, especially in the Chinese regions. The other products underestimated duration by only 1 to 2 units compared to the results calculated using ground measured precipitation. The severity of agricultural drought was estimated to varying degrees among the different selections. According to the observation results, the severity declined in the Chinese area from the middle to the side. The annual drought accumulation in the eastern Russian region was higher than in the northern ones. GPM and ERA5-Land had relatively better performances in terms of consistency with the gauge datasets in the Chinese section, while ERA5-Land showed high overestimation in the C-II region. Both ERA5 and MSWEP showed high overestimation of the referenced results in all regions, despite their excellent capture of duration.

The indicator, which can represent the ability to monitor heavy drought events, the general pattern of the max drought intensity comparison among the different selections, maintained a similar arrangement with the drought severities, while the variation scope among the corresponding records was relatively reduced. From the results provided by the observations, the range of intensities was between 0.9 and 1.6; the middle regions of the Chinese and eastern Russian regions tended to suffer much heavier drought events. The information on emergence times between 2015 and 2019 showed that March, June, and September were three common months in which highly affected events frequently occurred on the Russian side, while October experienced higher durations for Chinese regions. Considering the two aspects, including the intensity value and occurrence range, the products that had ideal performances were GPM (C-I and C-II), ERA5-Land (C-III), and MSWEP (R-I and R-II).

## 4. Discussion

### 4.1. Multi-View of the Strengths and Weaknesses of the PPs

In this study, four datasets were analyzed and their characteristics are presented in Table 1. GPM and MSWEP were found to have the highest spatial resolutions, but their weakness lies in their limited temporal spans and resolutions. Conversely, the ERA5 series was observed to provide relatively abundant information on temporal issues but lacked spatial resolution (0.25° in ERA5) and coverage (limited to land regions in ERA5-Land). Based on the above information, it can be concluded that reanalysis products are more suitable for long-term and large-scale research, while GPM and MSWEP are appropriate for contemporary precipitation evolution studies at medium scales.

In addition to the attribute information on the strengths and weaknesses of the products, this research analyzed consistency indices to explore potential links with different timing scales. On an annual scale, both regional and station-based analyses showed that ERA5-Land generally outperformed the other datasets and accurately estimated low precipitation levels at the stations. This finding aligns with previous research demonstrating the superior performance of reanalysis products in high-latitude regions with state-of-the-art assimilation technologies [24,25,27,44]. On a monthly scale, GPM estimation was recommended for the Chinese side, as it was fully calibrated by monthly gauge data before the final product release. However, the relatively insufficient meteorological monitoring information in the Russian Far East may affect the validity of this recommendation. MSWEP, with its optimal merging of estimates from multi-sourced datasets, has the potential to describe the corresponding results in such cases.

The study also revealed supporting findings on the spatial distribution and statistical metric evaluation at the daily scale, respectively. In the Chinese area, detailed comparisons of CC and BIAS values showed that ERA5-Land data were recommended for use in high-altitude regions, as previous research has also suggested [19,27,44], with the southeastern and northern regions with elevations greater than 1000 m being the main advantageous parts of the selection, while GPM yielded ideal responses for differentiating scores for RMSE, the index which represents the stability of the data being evaluated. Incorporating gauge-based data from the Climate Prediction Center (CPC) into the correction work, the introduction of global daily precipitation datasets enhanced the GPM capability to weaken the dispersion degree between estimation and observations. This finding is consistent with previous research conducted in the Chinese area [44–46]. Overall, the selection of appropriate datasets should be based on specific research requirements and the strengths and limitations of each dataset.

In terms of evaluating precipitation intensities, GPM demonstrated superior performance in mitigating the negative effects of underestimation, particularly during seasons such as spring and winter, when weather conditions can more easily affect the monitoring effectiveness of precipitation products (PPs). In summer, precipitation patterns typically peak. MSWEP has been shown to assemble relatively high-quality information from multiple sources of precipitation monitoring, resulting in better consistency with observations, particularly for higher ranges of precipitation intensities. During winter, however, MSWEP's performance becomes weaker, and moderate overestimation exists.

Looking at the agricultural regions from a different perspective, MESWP demonstrates generally good performance in terms of precipitation intensity frequencies on an annual scale due to its comprehensive integration of resources. In contrast, the ERA5 series exhibit relatively weak capturing abilities for both over- and underestimated light precipitation events, as well as moderate-to-heavy precipitation classes. These findings are consistent with those of earlier studies [2,47,48]. However, in terms of various frequency laws, there were no significant differences between the ERA series and other products. In some agricultural regions located in the eastern part of the Chinese area, the performances of ERA5 were superior for specific ranges in estimating extreme precipitation events.

When considering drought characterization, the analysis based on aSPI showed that the magnitudes and trends of results remained stable. MSWEP and GPM performed well

in both Chinese and Russian regions, while the opposite was found for PP estimation with ERA5. The adapted index, calculated based on MP, was consistent with the evaluation process of applicable analyses for the agroclimatic regions (as discussed in Section 3.1.2), and similar findings were reported in previous studies conducted in the NA region [44,49,50]. Based on the three elements that represent drought event characterization, ERA5 and MSWEP had an advantage in counting duration due to their short release latency. For the distribution of other indices, such as average severity and maximum intensity, the results based on GPM and measured precipitation were the most suitable for application in the Chinese region. When considering the best product for Russia, MSWEP showed relatively convincing responses through a comparative evaluation process of integration capability in the specific remote area.

### 4.2. Characteristic and Source Analysis of the Deviation from the Four PPs

Following the evaluation of the performances of the four precipitation products, it was necessary to conduct a detailed analysis of the objective and potential sources of the observed biases. This section discusses the comparison and correlation of the deviations among the selected products. Furthermore, it provides a comprehensive depiction of the unique characteristics of the reliable estimations and suggests potential improvements based on recent and current studies for future reference.

Apart from a few metrical indices and stationary results, we found that the performances of the MSWEP and GPM products in satellite observation technology were generally higher than those of reanalysis precipitation datasets of rain-gauge observations over the study area. However, the performances of all the types are not highly satisfactory at a daily scale. Therefore, further efforts are required to improve the accuracy of the PPs in practical applications. Nevertheless, from a meteorological perspective, the gaps between the state-of-the-art reanalysis PPs and the rest of the precipitation retrievals were relatively small. In some cases, ERA5-Land could be considered an alternative for hydro-meteorological applications at a large temporal scale. The PPs were evaluated against rain-gauge observations through "grid-point" methods; both the center-point differences between grids and inter-products and the deviation in spatial representativeness between the stationary data and the products may introduce errors in the evaluation results.

Despite the comprehensive merging of estimations from multi-satellites, errors and uncertainties in GPM estimation persist due to false positives in the identification of precipitation events caused by thick ice or snow cover in high-latitude regions during winter and spring and inaccurate estimations generated by IR measurements due to heavy clouds during widespread precipitation events. With this product, the data field named "precipitationCal" is recommended for original studies due to its maturity in merging different source materials with rain-gauge calibration. The working principles, including the method of interpolation and fusion processes through calculation from the sources mentioned above, may alleviate gaps in practical applications for sensitive areas. Another data field, "HQprecipitation", which merges all available MW sources, could be an effective alternative for the former field. In this research, based on daily selections from two types of records, we considered them as complementary information sources, as judged by gauge data, and enhanced the objectivity evaluation by comparing the corresponding results of other studies, especially with regard to the indices of RMSE and POD [9,21]. In future work, we encourage expanding the mechanism into a dynamic and automatic adaptation system to identify better results to represent GPM's monitoring level, rather than separating datasets into two fields. To estimate precipitation, Generative Adversarial Network-based precipitation estimation (precipGAN) was created using diverse data fields from this product. It provides an alternative algorithm that is accurate and computationally efficient, which can be implemented globally to produce satellite-based precipitation estimates [51].

Combined with the evaluation results obtained from agricultural drought indices and event characterizations, MSWEP and GPM are the optimal selections. Given the detailed advantage distribution of these two products, GPM would be suitable for monitoring

dry–wet conditions and displaying the severity and intensity of agricultural drought, particularly in most of the Chinese regions. On the other hand, the MSWEP exhibits powerful capacities in the respects mentioned when considering the Russian agricultural districts. In addition to these facts, the technology of near-real-time data release enables MSWEP to outperform ERA5 in event duration detection; the performance of ERA5 is inadequate in the other monitoring fields. Regarding the characterization of the different products in terms of drought monitoring, the abundant and large-cover calibration data sources enhance the ability to capture information for monitoring agricultural drought. With the overestimation in the range of "light" rain reported in the research on precipitation intensity identification, the reanalysis products tend to raise false alarms with the indices used to present the impact of disasters.

### 4.3. Research Limitations and Future Suggestions

This study has certain limitations that need to be addressed in future studies:

(1) There is a need to expand the range of products and potential applications used for evaluation. This study gives foundational information for sequential studies in the different zones. Since the optimal products for precipitation intensity and agricultural drought identification were confirmed, the eventual rules of extreme precipitation and drought in the specific area could be investigated by the corresponding products referenced in this study.

(2) The resolution mismatch and spatial attenuation of precipitation should be further solved. The resolution unification of the four products would ease the issue of large resolution differences between ERA5 and the other products, while interpolation with the bilinear method would introduce the other error in the area with complex terrain and water mass [27,52]. On the other hand, the spatial attenuation will result in the areal precipitation values being less than the observation values from the rain-gauge stations. Facing the above problems to be solved, we will search for more advanced downscaling methods and suitable areal reduction factors to enhance the accuracies of evaluation results.

(3) The duration of evaluating precipitation intensity needs to be reconsidered. In this study, 24 h was used as the scale of precipitation intensity based on the stipulation created by the CMA and the condition of the selected product. However, in certain circumstances, such as short-term heavy precipitation events, the distribution of precipitation per day can affect the determination of precipitation intensity. Therefore, smaller scales should be considered for monitoring extreme weather events, and the interval adjustment of precipitation intensity will be made dynamically based on the characteristics of different parts of the study area in future studies.

(4) Merging the multi-meteorological impacts needs to be further studied. The seasonal evaluation of the precipitation intensities implies that the monitoring conditions of the PPs varied with changes in the external environment. As the distinctive seasonal characteristics of the meteorological condition were analyzed, elements such as temperature and wind were discovered to be the primary factors influencing precipitation measurement and estimation [22,53]. Thus, dissecting the PPs' responses with respect to the evolution of these associated measurements deserves to be explored so as to clarify the change in the PPs' prediction accuracies with the development of natural conditions in different seasonal durations.

(5) Different agricultural drought degrees and indicators need to be further considered for evaluation. The characterization of drought events involved whole classes of agricultural drought occurrences, but varying degrees of drought will produce different effects on the environment. Therefore, the identification of different classes will be continued to monitor the efficacy of the products in capturing agricultural drought abilities. Moreover, with the recognition of drought diseases increasing, the evaluation of different precipitation-based indices, such as the Agricultural Precipitation Index (ARI) and the Soil Moisture Deficit Index (SMDI), should also be considered.

### 5. Conclusions

This study comprehensively evaluated reanalysis (ERA5 and ERA5-Land), satellite-based (GPM), and multi-source based (MSWEP) products against rain-gauge data over the Sino–Russian region located in the Amur River Basin at multiple scales from 2015 to 2019. The main conclusions include:

(1) Regarding spatial description, all four types of PPs show similar spatial distributions of annual mean precipitation, with a general increasing trend from west to east over the study area. For annual data consistencies at the site scale, ERA5 series data slightly outperformed other types in reducing gaps with observed data in the low-value range.

(2) In terms of consistency verification, GPM performed the best overall with the smallest RMSE, while MSWEP showed regional outperformance in the Russian region with the highest CC. The daily-assembled ERA5-Land data proved suitable for use in the Chinese area as a reanalysis resource. ERA5 performed lowest in most cases in this area, but its monthly data still deserve consideration in future studies of the eastern Chinese agroclimatic regions.

(3) When comparing the performances in identifying seasonal precipitation events, GPM can mainly alleviate intensity underestimation in summer, while MSWEP showed better quality in frequency simulation in this season. Both products also demonstrated superior identification accuracy and scoring indices among the different ATAs.

(4) In depicting agricultural drought conditions based on the distribution of the evaluated indices and derived event features, GPM and MSWEP showed clear regional characterization abilities in estimating conditions in the Chinese and Russian ATAs. However, reanalysis products were recommended to enhance their capacities by involving more empirical data regarding extreme precipitation events.

(5) The distinctive performances of the multi-typed precipitation products were analyzed systematically in terms of various aspects, and the optimal products were identified for each ATA. The results of this study not only offer guidance on the selection of PPs for local applications, but also have the potential to inform future improvements to existing products and R&D for the generation of new products.

**Supplementary Materials:** The following supporting information can be downloaded at: https://www.mdpi.com/article/10.3390/rs15102577/s1, Figure S1: Assessment of different precipitation intensities among the Chinese (a) and Russian (b) ATAs based on the POD and FAR; Figure S2: Spatial patterns of the aSPI-1 averaged by MP from 2015 to 2019 in the ATAs.; Table S1: The detailed information of selected station.

**Author Contributions:** Y.Z.: Conceptualization, Methodology, Formal analysis, Investigation, Data curation, Writing—original draft, Visualization. J.W.: Conceptualization, Writing—review and editing, Project administration, Funding acquisition, Supervision. E.G.: Conceptualization, Writing—review and editing, Supervision. K.L.: Investigation, Data curation, Writing—review and editing, Visualization. H.X.: Investigation, Data curation. All authors have read and agreed to the published version of the manuscript.

**Funding:** This study was funded by the Alliance of International Science Organizations (grant no. ANSO-CR-KP-2022-06), the Special Exchange Program of the Chinese Academy of Sciences (grant no. 2022000199), and the Construction Project of the China Knowledge Center for Engineering Sciences and Technology (grant no. CKCEST-2022-1-41).

**Acknowledgments:** We appreciate the anonymous reviewers for their constructive comments and insightful suggestions.

**Conflicts of Interest:** The authors declare that they have no known competing financial interests or personal relationships that could have appeared to influence the work reported in this paper.

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
