# Peer review of "Performance Evaluation of Multi-Typed Precipitation Products for Agricultural Research in the Amur River Basin over the Sino–Russian Border Region"

_remotesensing, doi:10.3390/rs15102577_

Round 1
Reviewer 1 Report
In this study, the author compared the error of ERA5, ERA5-LAND, MSWEP, GPM, and identify the agricultural drought using the PPs. Seasonal effects were found to be the main factors in exhibiting identification capabilities. The paper is interesting, but there are some suggestions. I recommend reconsideration of your manuscript following major revision.
1. In the introduction, the review of the drought index and agricultural drought identification was suggested to be added. It can be easy for readers to understand the innovation point of this paper.
2. Line 290: What is aSPI? Although the drought index was introduced in the reference, it was suggested to write the main formula of aSPI to ease readers’ understanding.
3. Line 293: The aSPI can be used to assess the drought impacts on vegetation. Why? Do you consider the effects of water demand on vegetation?
4. Line 301: In Figure 2, the author used disaster index as the axis name. What is the difference between the drought index and disaster index? Why did the author choose 0 as the threshold?
5. Line 545: In Figure 12, drought event characterization given by PPs and observations was shown. What is the observation? Is it the drought area in the yearbook? Do you explain and analyze the figure meaning in the paper?
Author Response
- In the introduction, the review of the drought index and agricultural drought identification was suggested to be added. It can be easy for readers to understand the innovation point of this paper.
Response 1: Thanks for this comment. The drought index and agricultural drought identification was added. Meanwhile, we reorganized the introduction to showing more related progresses about this topic.
- Line 290: What is aSPI? Although the drought index was introduced in the reference, it was suggested to write the main formula of aSPI to ease readers’ understanding.
Response 2: Thanks for this comment. The full name of aSPI is called “agriculture-oriented standardized precipitation index”. This index was modified based on the original standardized precipitation index (SPI) and the main formula of aSPI were put into the section 2.3.3.
- Line 293: The aSPI can be used to assess the drought impacts on vegetation. Why? Do you consider the effects of water demand on vegetation?
Response 3: Thanks for this comment. The reason for aSPI can assess the drought impacts on vegetation is that the index introduced the concept of effective precipitation (Pe) into the original SPI. The effective precipitation can be generally considered as the portion of the total precipitation that can be used productively by the plants, either directly or indirectly (without considering irrigation water). The aSPI takes account for total precipitation (P), crop evapotranspiration (ETc) and a soil water storage factor (SF). So, the water supply and demand related with vegetation (crop) is considered. We added the description of aSPI in the introduction and Section 2.3.3.
- Line 301: In Figure 2, the author used disaster index as the axis name. What is the difference between the drought index and disaster index? Why did the author choose 0 as the threshold?
Response 4: Thanks for this comment. The original Figure 2 is the schematic diagram of the run theory without real meaning. As for the question, the name on the axis were proposed that this theory can be afforded for multicable disaster with the statistical model (flood, drought etc.). As for the threshold, the value was set according to the classification of SPI value. For reduce the misunderstanding and improve the concise of the manuscript,we have deleted this figure and its description.
- Line 545: In Figure 12, drought event characterization given by PPs and observations was shown. What is the observation? Is it the drought area in the yearbook? Do you explain and analyze the figure meaning in the paper?
Response 5: Thanks for this comment. The results on the figure were produced as below: Firstly, the aSPI value are calculated by the precipitation values of the rain gauge station (observation) and PPs which obtained by the same location. Then the mean drought duration, severity and intensity were calculated based on the aSPI value and run theory. Actually, the “observation” in this figure is the corresponding results in point scale which were calculated based on the data collected by the station.

Reviewer 2 Report
This study presents a precipitation product evaluation study over a Northeastern region in China. The evaluated products are GPM-IMERG, ERA-5, and MSWEP. The study introduced a few extra analyses but the overall conclusions are much the same as other precipitation product studies that involve these three products. Therefore, I do not recommend this manuscript for publication.
Major issue:
There have been many precipitation product evaluation studies involving GPM, ERA-5, and MSWEP globally, regionally, and locally. First of all, they are not new precipitation products, 2015, 2020, and 2017 respectively. It is difficult to find the novelty of the study as the evaluation results follow other regional and global evaluation studies that I have read. I understand the study has a twist to add analysis for ATAs for heavy precipitation intensity and drought detection, but I don’t believe this is enough to set this study apart from countless previous studies. I would suggest conducting a specific agricultural study that uses agricultural data as an observation reference to analyze the best precipitation product.
The literature review is poor, which does not cover similar studies that have been conducted by other researchers.
Reviewer 3 Report
The paper compares a few precipitation data-sets in a region between Russia and China. The paper is interesting, but a few things have to be improved for the sake of the readers.
The paper is too long. I understand that using more data-sets can lead to a long paper, but I invite the authors to make the paper shorter.
Despite the length of the paper there is missing information.
Specifically at line 237 is not explained how the interpolation of ERA5 data from the 0.25° x 0.25° grid to 0.1° x 0.1° grid was made. Moreover, the authors do not discuss the implications of such interpolation. For example, because of different orography it is possible that such a difference would introduce an error when regridding the data. Another error can be obtained when using the bilinear interpolation that does not conserve the water mass (see for example Accadia et al. 2003 ).
It is not clear how the comparison between Pps and gauges was performed. Did the author interpolate Pps to rain gauges? Did the authors know that areal precipitation is generally less the point observation and a reduction factor is used to transform point measurement to areal value? Have the author applied any transformation to compare effectively point measurements with areal values?
During the paper the authors call ERA5 and ERA5-land data-sets as model data-sets, but they are not model data, but reanalysis. This means observations are elaborated in a model. The word "model" would be misleading and an unaware reader can think about a forecast model for example. I encourage the authors to replace the word "model" with the word "reanalysis".
The quality of some figures is very bad and should be improved.
Figure 1. It is difficult to see the region names C-1; C-2, etc
Figure 6 is blurry
Figures 9 and 10. Lines are too thin. It is very difficult to see the lines.
Figure 11 The legend is too small and the figure is confused and confusing. Please make it more readable.
Accadia, C., S. Mariani, M. Casaioli, A. Lavagnini, and A. Speranza, 2003: Sensitivity of precipitation forecast skill scores to bilinear interpolation and a simple nearest-neighbor average method on high-resolution verification grids. Wea. Forecasting, 18, 918–932
Reviewer 4 Report
Even though this is an interesting study, it could be useful for understanding the Precipitation Products for Agricultural Research and hydrological processes on Amur River Basin. However, the article needs some revisions. Please address the comments to improve the quality of your article.
1. Introduction needs improvement. For example, in introduction line number 46-47 authors mansion "... role in monitoring matter and energy exchanges and has a significant impact on both the atmosphere and land surfaces...", I recommend that the authors update this line to include the role of precipitation on energy exchanges on landscape and river networks and provide the following references: (a) Abed‐Elmdoust, Armaghan, Mohammad‐Ali Miri, and Arvind Singh. "Reorganization of river networks under changing spatiotemporal precipitation patterns: An optimal channel network approach." Water Resources Research 52.11 (2016): 8845-8860., (b) Abed-Elmdoust, Armaghan, Arvind Singh, and Zong-Liang Yang. "Emergent spectral properties of river network topology: An optimal channel network approach." Scientific reports 7.1 (2017): 11486, (c) Sarker, Shiblu. "Investigating topologic and geometric properties of synthetic and natural river networks under changing climate." (2021).
2. In table 1's Spatial resolution column, the resolution should be in "m" and not in degrees! I have doubts about the study area's projections! Please check twice.
3. There are numerous python toolboxes available. Including http://timcera.bitbucket.io/tsgettoolbox/docs/index.html. The data can be easily processed. Please reorganize section 2.2.2 with professionalism.
4. Table 2 is an established fact, so why do we require this?
5. Figures 1 require revision. On the map of their study area, authors may include river networks and systems. Please review the manual for ArcGIS or another professional software in order to generate publication-quality figures. Please refer to the following source. Gao, Y., Sarker, S., Sarker, T., & Leta, O. T. (2022). Analyzing the critical locations in response of constructed and planned dams on the Mekong River Basin for environmental integrity. Environmental Research Communications, 4(10), 101001.
6. Figures 2, 4, 7, 8, 9, 10, 11 and 12 require extensive revision. Authors may use Python or Matlab to create figures suitable for publication. Reduce the amount of white space between subplots.
7. Figures 3 and 5 need revision. Please review the manual for ArcGIS or another professional software in order to produce figures suitable for publication.
8. Please elaborate on figure 2 and how it relates to this study, as well as why this is significant.
9. This research does not persuade me that it provides a basis for environmental protection. What is the importance of this study? Please describe in a distinct section (prior to the conclusion) the potential implications of this study. Possibly revise section 4.3.
10. The references are not formatted appropriately!
Reviewer 5 Report
Reviewed manuscript “Performance Evaluation of Multi-typed Precipitation Products for Agricultural Research in the Amur River Basin over the Sino-Russian border region” is an original and interesting study. Authors comprehensively demonstrated/calculated the data of Agricultural Research in the Amur River Basin over the Sino-Russian border region. Study is very interesting and it would add scientific contribution in literature. Authors collected lot of data with strong reasoning. I would suggest minor revision.
Following are some suggestions for further improvements:
First few lines of the abstract should be about the importance of study area.
Lines 54-59: Consider revising the lines.
Lines 91-95: Please rewrite these lines.
Lines 164-174: Consider revising the lines.
Lines 355-363: please rewrite.
Introduction and Discussion section needs to further strengthen by latest studies on the subject. Improve the discussion section. Discussion is so long, please summarize it.
At some places in the text, there are grammatical mistakes that need to be corrected by some native English colleague. Please summarize the conclusion.
To further strengthen introduction, latest studies etc. are suggested to cite for Performance Evaluation of Multi-typed Precipitation Products for Agricultural Research or Agricultural Drought prone areas.
Round 2
Reviewer 1 Report
The author said, "the “observation” in this figure is the corresponding results in point scale which were calculated based on the data collected by the station." We can see that your study area is not a point. Therefore, how can you calculated the observation based on the data collected by the station?
Reviewer 3 Report
as far as I'm concerned I have no further comment.
Author Response
Thanks very much for your affirmation and support.
Reviewer 4 Report
Thanks for the revision.
Author Response
Thanks very much for your warm scrutiny.